# In What Languages are Generative Language Models the Most Formal? Analyzing Formality Distribution across Languages

**Asım Ersoy[1]\* , Gerson Vizcarra[2]\* , Tasmiah Tahsin Mayeesha[3]\*, Benjamin Muller[4]**

[1]Huawei Türkiye R&D Center    [2]Banco de Crédito e Inversiones
[3]North South University    [4]Sorbonne Université
asim.ersoy1@huawei.com gersonw.vizcarra@gmail.com
tasmiah.tahsin@northsouth.edu

## Abstract

Multilingual generative language models (LMs) are increasingly fluent in a large variety of languages. Trained on the concatenation of corpora in multiple languages, they enable powerful transfer from high-resource languages to low-resource ones. However, it is still unknown what cultural biases are induced in the predictions of these models. In this work, we focus on one language property highly influenced by culture: formality. We analyze the formality distributions of XGLM and BLOOM's predictions, two popular generative multilingual language models, in 5 languages. We classify 1,200 generations per language as formal, informal, or incohesive and measure the impact of the prompt formality on the predictions. Overall, we observe a diversity of behaviors across the models and languages. For instance, XGLM generates informal text in Arabic and Bengali when conditioned with informal prompts, much more than BLOOM. In addition, even though both models are highly biased toward the formal style when prompted neutrally, we find that the models generate a significant amount of informal predictions even when prompted with formal text. We release with this work 6,000 annotated samples, paving the way for future work on the formality of generative multilingual LMs.

## 1 Introduction

Natural Language Processing (NLP) systems are used worldwide across multiple cultures, audiences, contexts, communication goals, demographics, and languages. Thus, it is essential that these models be able to adapt to the sociocultural context of its users. As described by Hershcovich et al. (2022), linguistic style is one of the major dimensions by which cultures vary in NLP technologies.

In this work, we focus on formality. Formality is a stylistic property of language that can impact how we perceive a text. It typically carries information about the culture of the speaker (or writer), is constrained by the context of the message, and can impact the communicative goal of a text (Heylighen and Dewaele, 1999). Generating text with a desired

level of formality can be useful for different NLP applications (Hovy and Yang, 2021). For example, controlling the tone of machine translation models (Sennrich et al., 2016; Niu et al., 2017; Feely et al., 2019), designing chatbots with formality awareness to respond to user-preferred conversational style (Cox and Ooi, 2022), or assisting users to change the formality level of their writings (Rao and Tetreault, 2018; Wang et al., 2019, 2020).

Generative language models have demonstrated capabilities in producing cohesive texts and solving NLP tasks with zero/few-shot learning (Radford et al., 2019; Brown et al., 2020b; Chowdhery et al., 2022; Zhang et al., 2022), even in multilingual scenarios (Lin et al., 2021b; Scao et al., 2022; Barbieri et al., 2022; Jiang et al., 2022; Anil et al., 2023). Multilingual language models are trained with large amounts of text from different sources. That training process could make the model biased towards a certain level of formality because of the data of each language as well as cross-lingual transfer (Pires et al., 2019; Libovický et al., 2020; Muller et al., 2021), limiting the capabilities of the model to adapt to the different cultures of an NLP application.

This work analyzes the formality level of two multilingual language models: XGLM (Lin et al., 2021b) and BLOOM (Scao et al., 2022), across five languages, namely Arabic, Bengali, English, French, and Spanish. To do so, a native/proficient speaker of each language evaluates the generation outputs of each model into three categories: formal, informal, and incohesive. This evaluation allows us to analyze the generations across three different dimensions: the cohesiveness of the generations,[1] the formality bias given neutral prompts, and the formality preservation given formal/informal prompts. As an example, we show in Table 1 the predictions of BLOOM and XGLM conditioned on the same

---

\*Equal contribution. This work was done as part of the Fatima Fellowship mentoring program.

[1]In short, we define a sequence as incohesive if it cannot be evaluated as formal/informal. More details in Section 4.3

| Prompt | Generation | |
|---|---|---|
| | **XGLM(7.5B)** | **BLOOM(7.1)** |
| কাঁদতে না পারাটাও অনেক কষ্টের | কথা। কিন্তু আশা আর স্বপ্নের মানুষরা যে এত বদলায় ! এর চেয়ে বড় উদাহরণ আর হতে পারলো না ...! | !' এ কথা বললেও অঞ্জু ঘোষই সবচেয়ে বেশি কান্নায় ভেঙে পড়েছেন বলে জানিয়েছেন তার মেয়ে। গত বৃহস্পতিবার রাত ৯টার দিকে টঙ্গীর সরকারি হাসপাতালে মারা যান দেশের একজন বর্ষীয়ান অভিনেতা ও নাট্য ব্যক্তিত্ব আফজাল হোসেন। |
| *Not being able to cry is also very difficult* | *But people of our hopes and dreams change so much! There could not be a greater example...!* | *But Anju Ghosh was the one who broke down in tears, her daughter said. Anju Ghosh returned to work after spending time with her daughter. Afzal Hossain, a veteran actor and theater personality of the country, died at Tongi Government Hospital last Thursday around 9 pm.* |

Table 1: XGLM (Lin et al., 2021a) and BLOOM (Scao et al., 2022) generating predictions of different formality (informal in red and formal in green) based on a Bengali formal prompt sampled from the InFormal dataset (Krishna et al., 2022). As illustrated here, the formality of the predictions differs between XGLM and BLOOM.

prompt in Bengali but generating text of different formality level. Overall, our contributions are the following:

- To the best of our knowledge, this is the first work to analyze the formality of generative multilingual language models across multiple languages. While we have focused on specific models and languages in this work, the procedures followed to define formality, prompt sourcing, language generation, and measurement of how formality is preserved from prompts are generalizable to any generative system and language. We open-source 1,200 generations,[2] per language, manually annotated as formal, informal, or incohesive.

- We find that BLOOM generates about twice as long texts as XGLM. Besides, almost all the generated formal sentences are longer than the informal ones. Also, informal generations in English, French, and Spanish are characterized by being more conversational, and in Bengali, by having more punctuation marks.

- We find that BLOOM is significantly more cohesive than XGLM in English, French, and Spanish and performs similarly in other languages.

- Both XGLM and BLOOM are generally biased toward formal text when prompted in a

neutral way. However, both models are very sensitive to the formality of the prompt and will generate informal text if conditioned with an informal prompt. This is particularly striking for Arabic: BLOOM generates dialectal Arabic (considered informal) when prompted with informal text while being extremely biased toward Modern-Standard Arabic (considered formal).

## 2 Formality Across Different Languages

We start by defining formality in the five languages of our study.

**Arabic** The Arabic language is spoken in many dialects (Watson, 2011). These dialects are variants of classical or standard Arabic, which has a modernized version of it called Modern Standard Arabic (MSA). Badawi (1973), in his famous book "Mustawayat Al-arabiyya Al-muasira Fi Misr" *The levels of contemporary Arabic in Egypt*, presents a theory on the relationship between standard Arabic (*Fusha*) and vernacular Arabic (*Ammiya*) in Egypt. His theory describes the situation as a continuum with 5 major divisions: illiterate colloquial Arabic, educated colloquial Arabic, elevated colloquial Arabic, modern standard Arabic, and classical Arabic. The first three divisions are *Ammiya*, which is considered informal and not necessarily grammatically correct. The last two divisions are *Fusha*, which is considered formal. However, the definition of what is formal and what is informal could depend on the problem at hand; for example, in one

[2]https://github.com/asimokby/formality-bias-analysis

case, elevated colloquial Arabic could be considered formal, while illiterate colloquial Arabic as informal. In our work, we define formality for Arabic as follows: a piece of text is formal if it contains no words coming from any Arabic dialect which is not considered as *Fusha*, following (Badawi, 1973)'s definition of *Fusha*. For example, the following sentence: أين أقرب مسجد؟ (*where is the closest mosque?*) is formed of only *Fasih*, formal, words. Similarly, a piece of text is informal if it contains a word coming from any dialect and not *Fusha*. For example, فين أقرب مسجد؟ — (*where is the closest mosque?*) is informal because of the word فين — (*where*) which is Egyptian Arabic.

**Bengali** Bengali has a complex and elaborate system of using pronouns to express the degrees of familiarity and formality between the participants in a conversation (Das, 1968; Uddin, 2019). T-V distinction (Brown et al., 1960) or the contextual usage of pronouns to convey varying levels of formality, familiarity, and politeness, which is found in many Romance languages (French, Italian, Spanish, etc.), can also be seen in Bengali. Bengali follows a tripartite form of second-person pronouns: আপনি / Apni (formal) for respected elders and strangers, তুমি / Tumi (polite) for siblings/friends or familiar people and তুই / Tui (informal) for those who are younger, children or very close friends. The third person *he / she* can be translated to তিনি / Tini (formal) vs সে / Se (informal), which encodes two levels of formality – honorific and non-honorific. Bengali pronouns can encode numbers such as singular/plural, but the notion of formality is not changed by gender or numerical properties (David, 2015).

The following are other considerations of formality in Bengali: (i) Texts containing a high frequency of Sanskrit-originated words can be considered formal. Agglutination/Compound words can be considered more formal compared to their analytical or elaborated forms. For instance, the words মৃত্যুবরণ (formal) / মারা যাওয়া (informal) — *death* have the same meaning, but a different formality (Panda, 1992; Nagarajan, 2014; Ghosh et al., 2022). (ii) Bengali pronouns agree with the verb in levels of formality and there are formal and informal variations of the same verb (David, 2015; Sultana, 2016). For instance, verbs like *Give*, *Eat*, *Go* can be written as দাও, খাও, যাও (formal) or দে, খা, যা (informal) depending on the context. (iii) Among Bengali speakers in Bangladesh, regional dialects like Sylheti, Chakma, and Chittagonian are generally considered informal while classical Bengali dialect (Sādhubhāsā) or standardized Bengali dialect (Cholito vasha) is considered formal (Ray et al., 1966).

**English** Formality in English is commonly defined as the language style used in a given situation. A formal speech, for instance, has a very careful selection of pronunciation, words, and structure (Richards and Schmidt, 2013). Heylighen and Dewaele (1999) divide English formality into two dimensions: a *deep formality*, characterized by the understanding of the precise meaning, avoiding ambiguity; and a *surface formality* which focuses on the rigorous selection of manners. Some recent works focus on the latter to evaluate formality using the selection of words (Brooke et al., 2010) and discarding the topic (Pavlick and Tetreault, 2016). Following Liardét et al. (2019), we use the following rules below to evaluate cohesive English text as informal: (i) Presence of contractions, abbreviations, and colloquial expressions; (ii) presence of grammar infelicities, that is, unsuitable expressions, inconsistencies in writing, and misspellings; (iii) high occurrence of delexical verbs and phrasal verbs; and (iv) higher involvement of human participants and subjective judgments, such as opinions.

**French** Formality is typically classified in French into three classes: *soutenu*, *courant* and *familier* (Gadet, 2005; Beeching et al., 2009). The register *soutenu* is reserved for legal documents, literature, or when addressing someone we want to show particular respect (e.g., a judge). It usually involves addressing someone with the second person plural (called *vousvoîment*). The register *courant* corresponds to the one used in day-to-day life, for instance, when we talk to someone new which is typically neutral and includes few grammatical errors. The register *familier* is the one used with friends, or within a family circle. It usually involves addressing someone with the second singular person *tu*. It can include a large portion of grammatical errors. Finally, it can also include slang and insults in its most vulgar form. In this work, following what was done in the XFORMAL work (Briakou et al., 2021b), we classify generated text into two classes. Soutenu is associated with the formal class while *familier* and *courant* with the informal class.

**Spanish** Formality in Spanish is commonly described by the T-V distinctions in the singular

second-person pronoun derived from Latin. Specifically, there are two possible translations for the English pronoun "you": *tú* is considered informal, and *usted* is formal. Both pronouns have different conjugations. The formality in sentences that use the singular second person is easily recognizable.

In the case of the other pronouns, the first person is often considered less polite than the third one (Stewart, 2001). For that reason, the third person is commonly used in scientific texts (Salazar et al., 2013). Aside from the pronouns and their conjugations, according to Cépeda and Tavera (2007), a formal text in Spanish should accomplish other characteristics such as: (i) Having no typographical or grammatical errors. (ii) Being a set of sentences referring to the same topic. (iii) Being arranged in paragraphs and having a coherent correlation between ideas using appropriate connectors.

In our work, we check the presence of slang or offensive terms in a sequence to classify text as informal. Then, T/V distinction in sentences written using the second person defines the formality level. In a similar way, sentences written in the third person have a bigger probability of being classified as formal compared to the ones written in the first person. The final priority is the layout: paragraph-structured sequences are considered formal in more scenarios than conversational-structured ones.

## 3 Related Work

**Biases of Generative LMs**   Recent literature on Large Language Models (LLMs) demonstrated social bias and prejudice against minorities (Sheng et al., 2021; Blodgett et al., 2020; Bender et al., 2021; Bommasani et al., 2021; Liang et al., 2021) in terms of many categories including gender (Sun et al., 2019; Cao and Daumé III, 2020; Felkner et al., 2022), race (Davidson et al., 2019), religion (Abid et al., 2021; Malik et al., 2022), occupation, politics and disabilities which result in the production of damaging content. To create multilingual bias evaluation frameworks, it has been argued that careful curation of culturally aware datasets is needed (Talat et al., 2022).

Many papers have focused on measuring social biases and stereotypes against historically disadvantaged groups and counteracting them for a limited number of languages like English (Nadeem et al., 2021; Nangia et al., 2020; Barikeri et al., 2021; Smith et al., 2022), French (Névéol et al., 2022), Hindi (Malik et al., 2022), but similar work has not been done for low-resource languages like Bengali. To our knowledge, the evaluation of multilingual models for measuring cultural biases like formality has not been attempted so far.

**Formality Analysis**   Previous work in formality analysis has focused on formality classification (Heylighen and Dewaele, 1999; Abu Sheikha and Inkpen, 2010; Pavlick and Tetreault, 2016; Dementieva et al., 2022), formality style transfer in English (Rao and Tetreault, 2018; Wang et al., 2019, 2020; Czeresnia Etinger and Black, 2019; Madaan et al., 2020; Yao and Yu, 2021; Briakou et al., 2021a), and in the multilingual setting (Korotkova et al., 2019; Briakou et al., 2021b; Krishna et al., 2022). Formality-sensitive machine translation to control the generation of machine translation models to target formality has received attention in recent years (Sennrich et al., 2016; Niu et al., 2017; Feely et al., 2019; Viswanathan et al., 2020; Niu and Carpuat, 2020; Schioppa et al., 2021) and benchmark MT datasets and models have been published (Nadejde et al., 2022; Rippeth et al., 2022).

Recently, several datasets with formality annotations have been introduced in English (Lahiri, 2015; Pavlick and Tetreault, 2016; Rao and Tetreault, 2018). XFORMAL (Briakou et al., 2021b), TAOCD (Zaidan and Callison-Burch, 2011) and InFormal (Krishna et al., 2022) extended formality style transfer to the multilingual setting. In the following sections, we describe our experiments and results for different languages.

## 4 Experiments

We evaluate different dimensions of formality of the generations of two popular generative multilingual language models: XGLM (Lin et al., 2021b) and BLOOM (Scao et al., 2022), in five languages: Arabic, Bengali, English, Spanish, and French. We hypothesize that the influence of high-resource languages in the corpus can involve biases in the formality of the whole models. To see their behavior in different scenarios, we employ distinct variations of prompt lengths and formality. In addition, we tweak some parameters when generating to avoid incohesive outputs.

**XGLM**   (Lin et al., 2021b) is a multilingual language model trained with 500 billion tokens belonging to 30 languages of Common Crawl. XGLM has five sizes ranging from 564 million to 7.5 billion parameters.

BLOOM (Scao et al., 2022) is also a multilingual generative language model trained on around 341 billion tokens from a corpus of 59 languages (13 of them are programming ones) to democratize huge pre-trained language models. BLOOM was released in different sizes ranging from 560 million to 176 billion parameters.

XLGM and BLOOM are decoder-only transformers pre-trained on a similar set of languages with a comparable amount of data. We compare checkpoints of similar size (i.e. we compare XGLM 2.9B with BLOOM 3B and XGLM 7.5B and BLOOM 7.1B[3]). Regarding the proportion and data sources on which both models were trained, BLOOM was trained on a more varied set of domains than XGLM in spite of the XGLM corpus being larger. In addition, the BLOOM corpus has a more balanced distribution of the amount of data of the languages evaluated in this study. More details about the quantity and sources of both models can be found in Appendix C.

## 4.1 Prompting for Formality Evaluation

We employ two prompting strategies to condition the generation of the models. In that way, the behavior of the model in different scenarios can be assessed.

**Short Neutral Prompts** A short prompt is composed of up to three words to condition the language of the output without giving any context that could impact the formality level. That allows us to measure the models' tendency to produce a certain formality level with a neutral input. For the lexicon of each language, we pick a set of common words[4] (or a combination of them to avoid the confusion of languages when generating) that can be used in both formal and informal sentences.

**Long Informal/Formal Prompts** This set of prompts is composed of truncated sentences extracted from existing formal/informal sources. Using these prompts, we can verify how much the models preserve the formality level of their in-

put. The sources of the prompts include formality datasets such as GYAFC (Rao and Tetreault, 2018), XFORMAL (Briakou et al., 2021b), InFormal (Krishna et al., 2022). We also include datasets crawled from the web (Zaidan and Callison-Burch, 2011; Cañete, 2019) and informal songs (Muñoz, 2018).

Table 2 presents the details of which words/group of words we use as short prompts, and the dataset sources of the formal/informal prompts for each language.

| | Neutral[+] | Formal* | Informal* |
|---|---|---|---|
| **ar** | لما (When/Then), نعم (Yes), هناك (There), لولا (Unless), لو (If), من (From), عند (At/When), والله (I swear), في (In), لا (No) | TAOCD (Zaidan and Callison-Burch, 2011) | TAOCD (Zaidan and Callison-Burch, 2011) |
| **bn** | আমি (I), তার (His/Her), যদি (If), এটা (It), কী (What), কেন (Why), সে (He/She), আচ্ছা (OK), কিন্তু (But), তারা (They) | InFormal (Krishna et al., 2022) | InFormal + Microblog dataset (Chowdhury and Chowdhury, 2014) |
| **en** | *The, I, This, He, She, You, They, We, Do, There* | GYAFC (Rao and Tetreault, 2018) | GYAFC (Rao and Tetreault, 2018) |
| **fr** | C'est (It is), Ils (They), Elles (They), Il (He), elle (She), ce (This), Est-ce que (question), Ça (That), Ce (This), Deux (Two) | XFORMAL (Briakou et al., 2021b) | XFORMAL (Briakou et al., 2021b) |
| **es** | Por la (For the), Las (The), Los (The), Por el (For the), Con unos (With some), Por que la (Why the), Se ha (It had), Por su (Because of), Para un (For a), De una (Of a) | Wikipedia (Cañete, 2019) | 9322 rap lyrics in Spanish (filtered) (Muñoz, 2018) |

Table 2: Prompts used in our experiments. [+]List of the short prompts across the 5 languages. 10 prompts per language are used for 10 generations sampled for each prompt. *Sources of the formal/informal prompts. 100 prompts per language are sampled from these datasets.

## 4.2 Generation Parameters

Decoding parameters are essential because they can directly affect a language model's output. For each language, we select a set of parameters to produce fluent text that can be appropriately evaluated. All selections were chosen to impact the formality

---

[3]We use the checkpoints and implementations from https://huggingface.co/models

[4]http://corpus.rae.es/lfrecuencias.html, https://www.pinhok.com/kb/bengali/98/100-basic-bengali-vocabularies/, https://talkinarabic.com/arabic-words/, https://en.wikipedia.org/wiki/Most_common_words_in_English https://strommeninc.com/1000-most-common-french-words-frequency-vocabulary/

| Model/Language | Arabic | Bengali | English | French | Spanish |
|---|---|---|---|---|---|
| XGLM(2.9B) | 9.3% | 8.0% | 6.7% | 16.0% | 6.7% |
| BLOOM(3B) | 13.3% | 4.3% | 3.3% | 12.0% | 3.3% |
| XGLM(7.5B) | 8.7% | 5.0% | 10.0% | 18.0% | 7.7% |
| BLOOM(7.1B) | 12.3% | 6.3% | **3.7%\*** | **8.7%\*** | **2.7%\*** |

Table 3: % of incohesive samples out of the 1200 generated samples for each language (300 samples per model). Percentages are averaged across prompt types (400 neutral, 400 formal, and 400 informal prompts). Bolded values show that the corresponding model is significantly better according to a permutation-based statistical test with a p-value ≤5%.

level of models as little as possible. This subsection presents our list of generation parameters to reproduce our experiments.

**Global generation parameters**  We modified the decoding procedures to avoid very short sentences, code snippets, and outputs in other languages to produce an assessable amount of outputs with a significant length to be evaluated. The parameters are listed in Appendix A.

Regarding the total number of evaluated outputs, we generated three sets for each evaluated model and language: 100 with short prompts, 100 with formal prompts, and 100 with informal prompts. That resulted in 1200 generated outputs for each language.

**Language-specific generation parameters**  Before generating the sequences for formality evaluation, we tweaked some logit parameters for each language. All modifications were done to obtain more fluent sequences and reduce incohesive outputs such as ones with generation repetitions or non-understandable text. This process was done with a varied set of prompts regardless of length and formality level.

We use sampling to obtain the generation outputs for both models. Three specific parameters were set for both models: We set *top-k* to 50, which truncates the number of tokens to sample from. We set a high *top-p* (Holtzman et al., 2019) to generate diverse sampled tokens by cumulative frequency, and a high *temperature* (Ackley et al., 1985), which does not skew the distribution towards high probability tokens. The specific details of the parameters to reproduce our experiments can be found in Appendix A.

### 4.3 Formality Evaluation

We assessed the formality of all generated outputs. To do so, one native/proficient speaker of each language classified all 1200 generated sequences individually. We opted for this evaluation procedure because, at the time of performing the experiments, to our knowledge, there were no multilingual formality classifier models that included Arabic, Bengali, English, Spanish, and French. To avoid possible biases, each generated output was annotated without looking at its prompt and in a randomized order. All annotations were done by the authors themselves who are native speakers of the languages.

To validate the quality of our annotation process, we also collect 3 ratings (collecting the annotations from 2 extra annotators) per sample for 50 samples and report the observed inter-annotator agreement in the Appendix in Table 7. We find the inter-annotator agreement to be above 59% for all the languages reported.

The classification categories for all languages are **formal**, **informal**, and **incohesive**. A sequence is classified as formal or informal according to the rules of each language described in section 2. The "Incohesive" label is only assigned under certain conditions, such as sequences written in other languages, non-understandable text (with e.g. many repeated symbol/blank tokens) that cannot be evaluated for formality level, or code snippets.

## 5 Results & Analysis

We start by analyzing the cohesiveness of each model. We then focus on the cohesive text for analyzing formality.

### 5.1 Cohesiveness of Generation

As seen in Table 3, BLOOM(7.1B) generates significantly more cohesive texts than XGLM(7.5B) for English, French, and Spanish with p-values under 5%, of a permutation-based statistical test.

Interestingly, the results in Table 3 also show that a larger model does not necessarily lead to more cohesive generations. For example, BLOOM(3B)

| Model/Language | Arabic | Bengali | English | French | Spanish |
|---|---|---|---|---|---|
| XGLM(2.9B) | 92% | -3% | 14% | 41% | 58% |
| BLOOM(3B) | 100% | -6% | -6% | **-1%*** | 79% |
| XGLM(7.5B) | **83%*** | 33% | 8% | 32% | 45% |
| BLOOM(7.1B) | 100% | **-3%*** | -13% | 14% | 67% |

Table 4: % differences between formal and informal predictions (400 samples per language) sampled with neutral prompts. Light Gray indicates a bias toward formal generations and Dark Gray indicates a bias toward informal generations. Bolded values show that the corresponding model is significantly better (i.e. closer to 0) than the other model of the same size (based on a permutation-based test with p-value $\leq 5\%$).

generates more cohesive texts than BLOOM(7.1B) for Bengali and English. XGLM(2.9B) also generates more cohesive texts than XGLM(7.5B) for English, French, and Spanish. Is worth noting that we are only evaluating cohesiveness in a binary way (cohesive vs. incohesive) and are not judging the quality of the predictions beyond that.

Besides, the percentage of incohesive texts is noticeably higher for some languages than others for both BLOOM and XGLM. For example, the highest percentage of incohesive texts in the case of Bengali, English, and Spanish is less than or equal to 10%, while that percentage is higher in the case of Arabic and French.

## 5.2 Formality-Level Bias

Neutral prompts, given to an assumingly unbiased model, should lead to equal distributions of formal and informal generations with a difference close to zero between both generations. However, this is not the case here, as shown in Table 4. In the case of Bengali, we see that XGLM(2.9B), BLOOM(3B), and BLOOM(7.1B) are almost neutral with minor differences of -3% -6% and -3%, respectively, showing bias toward informal generations. On the other hand, we see XGLM(7.5B), surprisingly, showing significantly more bias toward formal generations than BLOOM(7.1B) with a difference of 33%. Upon qualitative analysis, we found that many of the generations of XGLM(7.5B) had Bengali religious Islamic text-like attributes that were considered formal during annotation, and the usage of hashtags or emojis was also less than the smaller model for neutral prompts.

BLOOM, for French, continues to show less bias showing only a bias of 1% toward informal generations in the case of BLOOM(3B) and 14% towards formal generations in the case of BLOOM(7.1B). On the other hand, XGLM(2.9B) shows significantly more bias than BLOOM(3B) toward formal

generations with a difference of 41%. For English, XGLM and BLOOM both show a small bias (in terms of percentages) towards different directions. XGLM(2.9B) and XGLM(7.5B) show bias towards formal generations by 14% and 8% respectively. However, BLOOM(3B) and BLOOM(7.1B) display bias towards informal generations by 6% and 13% respectively. After a careful review of the predictions, we find that French and English informal predictions of BLOOM are due to a large proportion of informal generated dialogs.

BLOOM, this time for Spanish, shows extreme bias towards the formal generations with a difference of 79% for BLOOM(3B) and 67% for BLOOM(7.1B). On the other hand, XGLM exhibits less bias towards formal generations with a difference of 58% for XGLM(2.9B) and 45% for XGLM(7.5B). These values indicate that both models are influenced by formal sources. In fact, most of the generated sequences with short prompts have the style of news and Wikipedia articles.

A biased distribution of outputs could be reasoned by the data the model was trained on. As stated in BLOOM (Scao et al., 2022), the biggest part of the corpus for Arabic was the Arabic-focused Masader repository (Alyafeai et al., 2021; Altaher et al., 2022), which is dominated by Modern Standard Arabic (MSA) that is considered formal (cf. section 2). This explains the extreme bias BLOOM(3B) and BLOOM(7.1B) show towards formal generations with a bias of 100%. XGLM(7.5B) similarly shows an extreme bias toward formal generations, but significantly less than BLOOM(7.1B) with a difference of 83%.

In terms of model size, we notice that XGLM(2.9B) shows more bias towards formal or informal generations than XGLM(7.5B) for all the languages except Bengali, which could indicate that the bigger the XGLM model's size, the less biased it is. On the other hand, this isn't the case for

| Model/Language | Arabic F→F% / I→I% | Bengali F→F % / I→I% | English F→F% / I→I% | French F→F% / I→I% | Spanish F→F% / I→I% |
|---|---|---|---|---|---|
| XGLM(2.9B) | 89.4% / 61.1% | 79.8% / **100.0%**\* | 34.0% / 94.0% | 26.7% / 59.5% | 85.9% / 80.2% |
| BLOOM(3B) | 94.2% / 55.1% | 83.7% / 87.1% | 29.2% / 91.7% | 32.0% / **82.0%**\* | 77.8% / 90.4% |
| XGLM(7.5B) | 88.6% / **76.7%**\* | 75.5% / 98.8% | 34.4% / 84.7% | **54.0%**\* / 75.6% | 86.9% / 75.8% |
| BLOOM(7.1B) | 93.5% / 51.1% | 74.0% / 91.9% | 27.6% / **94.0%**\* | 25.8% / 66.7% | 83.8% / **96.8%**\* |

Table 5: Formality preservation samples' percentages for **Formal / Informal** prompts (800 prompts per language: 400 formal and 400 informal). Each sample is annotated as either formal, informal, or incohesive and the percentages are calculated without incohesive text counts. Bolded values show that the corresponding model is significantly better according to a permutation-based statistical test with a p-value $\leq 5\%$.

BLOOM as BLOOM(3B) is only expressing more bias for Bengali and Spanish, while BLOOM(7.1B) shows more bias for English and French.

In summary, the models show moderate bias for some languages such as English and Bengali, except for XGLM(7.5B) in the case of Bengali, while also showing extreme bias for other languages such as Arabic, French, and Spanish. This difference might be caused by the fact that every language is present in the data with a different percentage and is coming from different sources as shown in Table 8. Overall, it is noticeable that the bias is mostly toward formal generations for all the models and for all the languages.

### 5.3 Formality-Level Preservation

In this experiment, we measure how well the formality of a generation is the same as the formality level of the prompt (i.e. how well the model preserves the formality-level of the prompt). We find that the formality style of the prompts is preserved efficiently for some languages by some models while being almost ignored in some other cases.

For Arabic, as we show in Table 5, BLOOM(3B) and BLOOM(7.1B) preserve the formality style of 94.2% and 93.5%, respectively, of the samples when the given prompt is formal. We note that despite being highly biased toward formal text in Arabic (as seen in section 5.2), both models are able to preserve the style of informal prompts, at least 51.1% of the time.

XGLM(2.9B), for Bengali, preserves the style of the informal prompts of significantly more samples than BLOOM(3B) with a percentage of 100%. BLOOM pays attention to the informal style of the prompts as well, unlike the case for Arabic, and preserves the style of 87.1% of the samples generated with BLOOM(3B) and 91.9% of the samples generated with BLOOM(7.1B).

Both BLOOM and XGLM, this time for English, do not preserve the formal style of the prompts for more than 34.4% of the samples for any model. However, they both preserve the informal style in at least 84.7% of the generated samples with BLOOM(7.1B) preserving significantly more samples than XGLM(7.5B). A similar trend follows for French with both BLOOM and XGLM unable to preserve the formal style for more than 32.0% of the samples in the case of XGLM(2.9B), BLOOM(3B) and BLOOM(7.1B). On the other hand, XGLM(7.5) preserves the formal style significantly better than BLOOM(7.1B) with a percentage of 54.0%. And again the informal style is being preserved better with, specifically, BLOOM(3B) which preserves the style better than XGLM(2.9B) with a percentage of 82%.

The formal and informal styles in Spanish are preserved consistently across the models to at least 77.8% of the samples with formal prompts and at least 75.8% with informal prompts with BLOOM(7.1B) preserving the style in significantly more samples than XGLM(7.5B).

In terms of model size, we notice that the size of the model is not an indicator of how well the model can preserve the formality style. For example, BLOOM(3B) preserves the formal style better than BLOOM(7.1B) for all languages except Spanish. In summary, we see that the informal style is mostly preserved well for most languages except with BLOOM for Arabic. The formal style, on the other hand, is mostly preserved well for all languages except English and French.

### 5.4 Typographic and Lexical Differences between Formal/Informal Generations

We report in Table 9 general statistics about the generated texts of each model and language by formality level. Results show that BLOOM gener-

ates about twice longer texts as XGLM. In terms of the average number of sentences per generation, BLOOM, when the generation is informal, generates more and shorter sentences than when the generation is formal. Also, informal generations tend to have emojis as expected, especially in the case of Bengali. Besides, informal generations tend to have more punctuation marks than formal ones. Finally, the results of the average number of new lines and the average number of "-", which are used to signal dialogues, support what we mentioned earlier about BLOOM's tendency to generate conversational text.

## 6 Discussion

Formality bias when present in multilingual models, which are increasingly popular nowadays, can lead to undesirable outcomes. For example, using *"please"* is common among North American English native speakers in requests, even among close friends, while in Arabic, it could be considered awkward, if not rude, in conversations among close friends (Hovy and Yang, 2021). A usage example of language models is solving downstream tasks using prompting techniques for zero-shot learning, such as (Zhong et al., 2021)'s work on question-answering. Prompting has also been used with large language models for conversational chatbots such as ChatGPT (Ouyang et al., 2022). As prompting is becoming popular, we must understand that prompting a model that exhibits formality bias could be a barrier to getting the expected output. Furthermore, depending on the application, formality bias could even lead to sometimes unwanted misunderstandings (Hershcovich et al., 2022) and conflicts if the models, for example, are not able to generate text in the formality style of the users' expectations.

Controlling LLMs generations has been taken into consideration in recent work, such as in (Ouyang et al., 2022), where they fine-tune a language model (Brown et al., 2020a) intending to align the model with the intent of the users using reinforcement learning from human feedback (RLHF) (Christiano et al., 2017; Stiennon et al., 2020). Future work could analyze the impact of RLHF on the formality distributions present in language models. Furthermore, our work focused only on two pre-trained models with up to 7B parameters. The same analysis could be conducted for larger models such as GPT-3 and BLOOM(175B).

Finally, the increase in the number of multilingual language models calls for more work on their biases.

## 7 Conclusion

In conclusion, we analyzed the formality level of the generations of two large-scale generative language models, XGLM and BLOOM, ranging from 2B parameters to 7B parameters. We first observed the cohesiveness of the predictions. We found that BLOOM(7.1B) predicts significantly more cohesive text than XGLM(7.5B) for English, French, and Spanish. Second, we showed that, across all five languages, both models tend to generate formal text when prompted neutrally. Finally, we found that the formality of the prompt highly impacts both models. In most cases, they generate the same style as the prompt, with slight differences between the models depending on the language. Our analysis is based on the annotations of 1,200 generations in Arabic, Bengali, English, French, and Spanish. We release them with this paper opening future avenues for modeling the formality of generative multilingual language models.

## 8 Limits

In this work, we experiment only with two models, XGLM and BLOOM. The limitation of computing resources tied us with models that have 7.5B parameters or less and the limitation of financial resources and manpower held us back from experimenting with more languages. Furthermore, the generated samples of each language were annotated by only one annotator, except for English. To validate our methodology we asked two extra native speakers of each language to annotate 50 samples per language. We share the observed inter-annotator agreement values in Table 7 (Gwet, 2014). We find the observed agreement to be above 59% for all the reported languages, supporting the quality of our annotation process. Finally, we note that despite the aforementioned limitations, the methodology used in this study can be applied to any generative system and language.

## 9 Acknowledgment

We thank the Fatima Fellowship[5] and Hugging Face for organizing and sponsoring the Fatima Research Fellowship program.

---

[5]cf. https://www.fatimafellowship.com/

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

## A Generation parameters

We list here the decoding parameters for both models:

1. We filter out the generation sequences that are not natural language (i.e., code) by excluding from the generation process all the tokens that contain any of the following symbols: {, }, (, ), [, ], \\, <, >, |, and ; \n.

2. We force the model to generate at least 30 new subword tokens (excluding the prompt) to have a long enough generation sequence and be able to assess formality.

3. We set a maximum of 150 new tokens of generation to avoid long outputs that could include multiple formality variations.

4. Length of the prompts. For the short-prompt setting, we employ at most three tokens to condition the generation in the desired language. For the formal/informal prompts, we use 15 words (based on white-space tokenization) on average.

5. Table 6 shows the language-specific generation parameters we used for both BLOOM and XGLM.

|  | Top-k | Top-p | Temperature |
|---|---|---|---|
| Arabic | 50 | 0.95 | 1 |
| Bengali | 50 | 0.95 | 1 |
| English | 50 | 0.95 | 1 |
| French | 50 | 1 | 0.8 |
| Spanish | 50 | 1 | 0.8 |

Table 6: Language-specific generation parameters for both models

|  | Observed Agreement |
|---|---|
| Arabic | 0.62 |
| Bengali | 0.62 |
| English | 0.79 |
| Spanish | 0.59 |

Table 7: The inter-annotator agreement (IAA) values per language. The values, representing the observed agreement, are based on the annotations, of a sample of data, from 3 different native annotators of each language for 50 random samples.

## B Descriptive statistics of the generations

General statistics of the generations are in Table 9 reported per language for each model and generation label pair. The table contains the following statistics: the average length of the generation, the average number of sentences in a generation, the average length of the sentences, the average number of emojis per generation, the average number of punctuation marks per generation, the average number of new lines per generation, and finally, the average number of the dialogue mark/dash (-) per generation.

## C XGLM and BLOOM training corpora

We show in Table 8 details of the languages used in our analysis in the training corpus of BLOOM and XGLM.

## D Formality Distribution

We visualize the annotated data for each language to help in seeing an overview of all the results. Each language is represented by a plot, see Figures 1, 2, 3, 4, and 5, with 12 bars with 3 bars corresponding to each model representing the 3 prompts types: formal informal, and neutral. Each bar in the plot represents 100 texts generated with the corresponding model when prompted with the corresponding prompt type. The colors in each bar represent the 3 possible annotations: formal, informal, and incohesive.

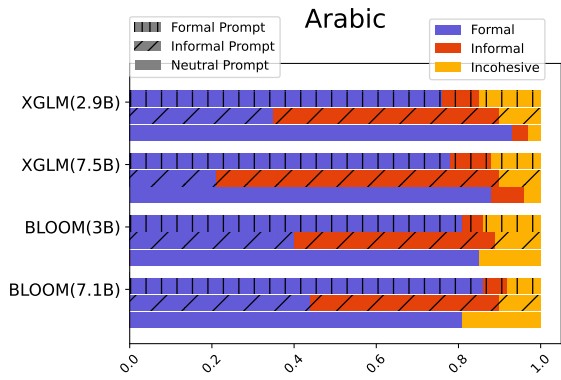

Figure 1: Plot of the distribution of the generations for Arabic, for each prompt type, according to their labeled categories: Formal, Informal, and Incohesive. Each bar in the plot represents 100 generations.

| | Corpus Size GiB (%) | | Data source domains | |
| --- | --- | --- | --- | --- |
| | **XGLM** | **BLOOM** | **XGLM** | **BLOOM** |
| **ar** | 64.34* (0.88%) ~92 upsampled | 69.71 (4.34%) | Web | Web, news, books, subtitles, Wikipedia, wikisources |
| **bn** | 11.19* (0.15%) ~76 upsampled | 17.32 (1.15%) | Web | Web, Wikipedia, Wikisource, open-source NLP datasets |
| **en** | 3,324.45 (45.66%) | 451.64 (30.04%) | Web | Papers, Web, patents, books, subtitles, forums, Wikipedia, news |
| **fr** | 303.76 (4.17%) | 193.94 (12.90%) | Web | Web, scholarly documents from all academic fields (HAL), Wikisource, Wikipedia, subtitles |
| **es** | 363.83 (4.99%) | 163.07 (10.84%) | Web | Web, subtitles, Wikipedia, news, magazines |

Table 8: XGLM (Lin et al., 2021b) and BLOOM (Scao et al., 2022) quantity and data sources in pre-training corpus. XGLM used upsampling for some languages including Arabic and Bengali.

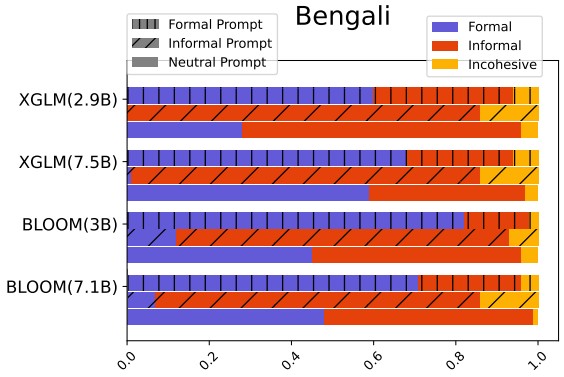

Figure 2: Plot of the distribution of the generations for Bengali, for each prompt type, according to their labeled categories: Formal, Informal, and Incohesive. Each bar in the plot represents 100 generations.

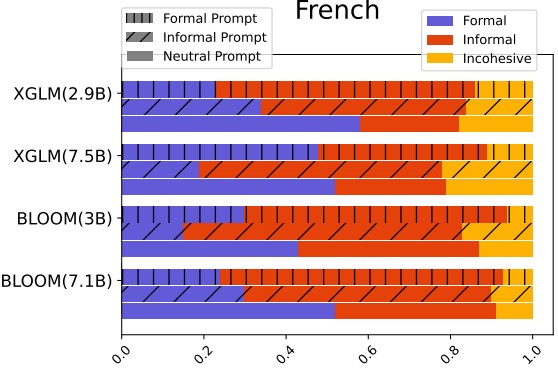

Figure 4: Plot of the distribution of the generations for French, for each prompt type, according to their labeled categories: Formal, Informal, and Incohesive. Each bar in the plot represents 100 generations.

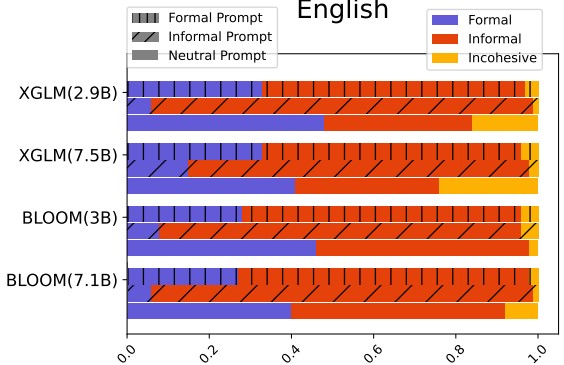

Figure 3: Plot of the distribution of the generations for English, for each prompt type, according to their labeled categories: Formal, Informal, and Incohesive. Each bar in the plot represents 100 generations.

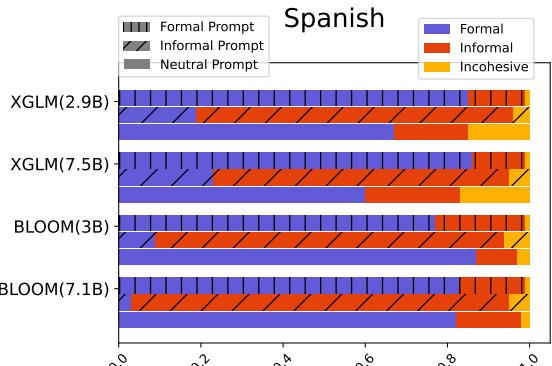

Figure 5: Plot of the distribution of the generations for Spanish, for each prompt type, according to their labeled categories: Formal, Informal, and Incohesive. Each bar in the plot represents 100 generations.

**Arabic**

| Prompt/ Statistic | Avg. Length | Avg. # of sentences | Avg. Length of sentences | Avg. # of emojis | Avg. # of punctuation marks | Avg. # of new lines | Avg. # of dialogue mark(-) |
|---|---|---|---|---|---|---|---|
| $XGLM(2.9B) - Informal$ | 175.074 | 1.338 | 130.593 | 0.000 | 6.794 | 0.000 | 0.000 |
| $XGLM(2.9B) - Formal$ | 246.696 | 1.686 | 145.895 | 0.010 | 4.755 | 0.000 | 0.000 |
| $BLOOM(3B) - Informal$ | 446.444 | 2.037 | 218.664 | 0.000 | 5.463 | 2.185 | 0.019 |
| $BLOOM(3B) - Formal$ | 495.403 | 3.583 | 137.523 | 0.005 | 6.820 | 3.626 | 0.345 |
| $XGLM(7.5B) - Informal$ | 187.345 | 1.471 | 127.023 | 0.000 | 9.391 | 0.000 | 0.000 |
| $XGLM(7.5B) - Formal$ | 244.610 | 1.620 | 150.581 | 0.000 | 3.299 | 0.000 | 0.000 |
| $BLOOM(7.1B) - Informal$ | 441.538 | 2.692 | 163.357 | 0.000 | 5.404 | 2.019 | 0.058 |
| $BLOOM(7.1B) - Formal$ | 506.123 | 3.185 | 158.176 | 0.000 | 5.905 | 2.645 | 0.104 |

**Bengali**

| Prompt/ Statistic | Avg. Length | Avg. # of sent. per gen. | Avg. Length of sent. | Avg. # of emojis per gen. | Avg. # of punctuation marks per gen. | Avg. # of new lines per gen. | Avg. # of dialogue mark(-) |
|---|---|---|---|---|---|---|---|
| $XGLM(2.9B) - Informal$ | 151.734 | 1.552 | 97.431 | 0.357 | 17.487 | 0.000 | 0.000 |
| $XGLM(2.9B) - Formal$ | 164.149 | 1.256 | 130.467 | 0.000 | 3.091 | 0.000 | 0.000 |
| $BLOOM(3B) - Informal$ | 413.338 | 2.128 | 193.667 | 0.128 | 11.047 | 1.561 | 0.020 |
| $BLOOM(3B) - Formal$ | 384.252 | 1.338 | 286.909 | 0.014 | 5.518 | 1.288 | 0.007 |
| $XGLM(7.5B) - Informal$ | 167.110 | 1.728 | 96.294 | 0.360 | 15.507 | 0.000 | 0.000 |
| $XGLM(7.5B) - Formal$ | 152.767 | 1.248 | 122.199 | 0.008 | 2.880 | 0.000 | 0.000 |
| $BLOOM(7.1B) - Informal$ | 419.400 | 1.845 | 226.829 | 0.187 | 13.484 | 2.058 | 0.155 |
| $BLOOM(7.1B) - Formal$ | 418.500 | 1.198 | 349.046 | 0.000 | 5.063 | 1.127 | 0.008 |

**English**

| Prompt/ Statistic | Avg. Length | Avg. # of sent. per gen. | Avg. Length of sent. | Avg. # of emojis per gen. | Avg. # of punctuation marks per gen. | Avg. # of new lines per gen. | Avg. # of dialogue mark(-) |
|---|---|---|---|---|---|---|---|
| $XGLM(2.9B) - Informal$ | 225.720 | 3.332 | 67.047 | 0.005 | 7.518 | 0.000 | 0.000 |
| $XGLM(2.9B) - Formal$ | 261.529 | 3.103 | 83.544 | 0.000 | 5.943 | 0.000 | 0.000 |
| $BLOOM(3B) - Informal$ | 584.288 | 10.236 | 56.152 | 0.014 | 19.803 | 6.159 | 0.620 |
| $BLOOM(3B) - Formal$ | 646.354 | 6.829 | 93.727 | 0.000 | 12.537 | 2.159 | 0.000 |
| $XGLM(7.5B) - Informal$ | 241.613 | 3.497 | 68.359 | 0.022 | 8.680 | 0.000 | 0.006 |
| $XGLM(7.5B) - Formal$ | 281.921 | 3.371 | 82.887 | 0.000 | 6.000 | 0.000 | 0.000 |
| $BLOOM(7.1B) - Informal$ | 575.236 | 10.718 | 52.733 | 0.005 | 22.278 | 7.204 | 1.324 |
| $BLOOM(7.1B) - Formal$ | 639.466 | 6.808 | 93.020 | 0.027 | 14.123 | 2.959 | 0.110 |

**French**

| Prompt/ Statistic | Avg. Length | Avg. # of sent. per gen. | Avg. Length of sent. | Avg. # of emojis per gen. | Avg. # of punctuation marks per gen. | Avg. # of new lines per gen. | Avg. # of dialogue mark(-) |
|---|---|---|---|---|---|---|---|
| $XGLM(2.9B) - Informal$ | 207.861 | 2.723 | 75.713 | 0.058 | 8.927 | 0.000 | 0.000 |
| $XGLM(2.9B) - Formal$ | 231.435 | 2.652 | 86.646 | 0.000 | 6.861 | 0.000 | 0.000 |
| $BLOOM(3B) - Informal$ | 621.216 | 11.869 | 51.417 | 0.006 | 25.562 | 8.273 | 1.051 |
| $BLOOM(3B) - Formal$ | 612.727 | 6.125 | 99.197 | 0.000 | 13.909 | 2.205 | 0.034 |
| $XGLM(7.5B) - Informal$ | 208.567 | 2.850 | 72.525 | 0.047 | 9.323 | 0.000 | 0.000 |
| $XGLM(7.5B) - Formal$ | 235.277 | 2.891 | 80.735 | 0.000 | 7.403 | 0.000 | 0.000 |
| $BLOOM(7.1B) - Informal$ | 588.804 | 13.375 | 43.091 | 0.006 | 29.667 | 10.583 | 2.607 |
| $BLOOM(7.1B) - Formal$ | 637.415 | 6.500 | 97.216 | 0.000 | 15.679 | 2.425 | 0.066 |

**Spanish**

| Prompt/ Statistic | Avg. Length | Avg. # of sent. per gen. | Avg. Length of sent. | Avg. # of emojis per gen. | Avg. # of punctuation marks per gen. | Avg. # of new lines per gen. | Avg. # of dialogue mark (-) |
|---|---|---|---|---|---|---|---|
| $XGLM(2.9B) - Informal$ | 222.789 | 2.798 | 78.974 | 0.028 | 9.514 | 0.000 | 0.000 |
| $XGLM(2.9B) - Formal$ | 249.69 | 2.228 | 111.517 | 0.000 | 6.123 | 0.000 | 0.000 |
| $BLOOM(3B) - Informal$ | 553.59 | 9.291 | 58.689 | 0.000 | 21.000 | 5.427 | 0.846 |
| $BLOOM(3B) - Formal$ | 613.827 | 4.532 | 134.672 | 0.000 | 12.012 | 1.734 | 0.012 |
| $XGLM(7.5B) - Informal$ | 225.454 | 2.981 | 74.957 | 0.019 | 9.870 | 0.000 | 0.000 |
| $XGLM(7.5B) - Formal$ | 248.728 | 2.32 | 106.663 | 0.006 | 6.254 | 0.000 | 0.000 |
| $BLOOM(7.1B) - Informal$ | 530.218 | 8.589 | 60.846 | 0.000 | 20.565 | 5.435 | 1.331 |
| $BLOOM(7.1B) - Formal$ | 640.643 | 4.661 | 136.668 | 0.000 | 12.393 | 1.655 | 0.012 |

Table 9: The average statistics per formal and informal generation of sequence length, number of sentences, length of sentences, number of emojis, number of punctuation marks, number of new lines, and number of the dialogue mark ("-").