# OpenReview forum: "In What Languages are Generative Language Models the Most Formal? Analyzing Formality Distribution across Languages"
_EMNLP/2023/Conference — EMNLP 2023 Findings_

### Official Review · Reviewer_4rv1 · 2023-07-27

**Soundness:** 4

**Excitement:**

3: Ambivalent: It has merits (e.g., it reports state-of-the-art results, the idea is nice), but there are key weaknesses (e.g., it describes incremental work), and it can significantly benefit from another round of revision. However, I won't object to accepting it if my co-reviewers champion it.

**Missing References:**

I would expect a more detailed approach to annotation. Possible references:
https://aclanthology.org/2023.acl-long.697/ What’s the Meaning of Superhuman Performance in Today’s NLU? ACL 2023
https://aclanthology.org/P14-2083.pdf Linguistically debatable or just plain wrong? ACL 2014


**Paper Topic And Main Contributions:**

Τhe authors prompt two large language models (LLMs), namely XGLM and BLOOM, with neutral, formal and informal strings in five languages and evaluate their responses for cohesion, formality and informality. They use neutral prompts and non-neutral ones: the latter are pieces of text previously characterised as +/-formal.  The authors also check the impact of (i) the model's size and (ii) the type of texts on which the model has been trained on  the statistics of the type of prompt-type of response relation. The authors base their results on well-planned, substantial experiments.

**Reasons To Accept:**

Experiments are well-designed to check clearly specified variables. The results are based on a reasonable number of experiments.
Diagnostics of formality are particular to each language, which is linguistically correct.

**Reasons To Reject:**

Annotation of results would be more reliable if it had been done by more than 2 annotators for each language. The authors should have provided a profile of the annotators (age, education, social class). In fact, annotation has been performed in a rather lean way.

**Reproducibility:**

4: Could mostly reproduce the results, but there may be some variation because of sample variance or minor variations in their interpretation of the protocol or method.

**Reviewer Confidence:**

4: Quite sure. I tried to check the important points carefully. It's unlikely, though conceivable, that I missed something that should affect my ratings.

**Typos Grammar Style And Presentation Improvements:**

Presentation and language satisfactory.

655. which-->who

---

> ### Author Rebuttal · Authors · 2023-08-28
>
> Thank you for your positive review and helpful comments!
>
> We agree with your feedback that collecting at least 3 ratings per sample would ensure the quality of our annotation process. This project was done in a resource constraints setting, and we were not able to onboard more annotators. Following your suggestion, we will collect 2 more ratings for at least 50 samples that we will add to the next version of our paper. We will also report demographics information of the raters.

---

### Official Review · Reviewer_13h5 · 2023-08-04

**Soundness:** 3

**Excitement:**

4: Strong: This paper deepens the understanding of some phenomenon or lowers the barriers to an existing research direction.

**Paper Topic And Main Contributions:**

This evaluates the generations of two multilingual generative models for their level of formality. Constructing “neutral” prompts and using a selection of formal and informal text datasets, it evaluates to what extent models generate formal text “by default” in different languages, and whether they are able to “preserve” the given formality level.

**Questions For The Authors:**

Isn’t the “incohesive” label conflating nonsense generations and, e.g., outputs that are just too short to tell the formality level? Maybe it would make more sense to choose a more neutral term such as “unknown”, or differentiate the two cases?

**Reasons To Accept:**

- This is an interesting aspect of multilingual models and cross-lingual differences!
- There is a clear effort to operationalise formality levels, and to explain conditions in each language.
- The setup and evaluation is well-described, and the generated & annotated samples are released.

**Reasons To Reject:**

- The definitions of “informality” are not really consistent across languages. I understand that there are differences between languages, but for instance: How is soutenu/courant/familier actually defined? What does “deep formality” mean and why is it only mentioned for English? Is dialogue in English/languages without T/V distinction implicitly considered informal? If so, why?
- There is little attempt made to explain why the “formal” prompts in many cases (EN, most AR, FR, some ES) yield fewer formal continuations than the “neutral” ones. This looks like a calibration issue to me.

**Reproducibility:**

4: Could mostly reproduce the results, but there may be some variation because of sample variance or minor variations in their interpretation of the protocol or method.

**Reviewer Confidence:**

5: Positive that my evaluation is correct. I read the paper very carefully and I am very familiar with related work.

**Typos Grammar Style And Presentation Improvements:**

- l.158 German is listed as a Romance language
- l.389 you mean Appendix A, right?
- terminology: be careful not to confuse slang with grammar errors
- I would suggest using different cell background colours than green and pink, both because these somehow imply that more formal language is “better”, and because I doubt these are colourblind-safe.

---

> ### Author Rebuttal · Authors · 2023-08-28
>
> Thank you for your positive review and insightful comments.
>
> About your point on defining informality: after a careful literature review and discussions with linguists, our take for this work was that creating a consistent definition of informality was not feasible given the language- and culture-specific aspect of formality. We, therefore, chose to define formality carefully for each language independently. We agree that the mention of deep formality in English and the different standard registers in French requires more detailed explanations. We will improve Section 2 based on your feedback.
>
> Regarding the formality of generations given neutral vs. formal prompts. The percentages in Table 3 and Table 4 cannot be compared directly. The percentages in Table 3 represent the difference between the percentage of formal and informal outputs. On the other hand, the percentages in Table 4 represent, in the case of F->F, the percentages of formal outputs. However, upon a closer look, we indeed find that for English and French, neutral prompts lead to more formal outputs compared to formal prompt predictions (44%  to 30% for English, and 51% to 31% for French, on average). The formal output proportions for Arabic are more balanced between neutral and formal prompts. Finally, for Spanish and Bengali, formal prompts result in more formal outputs compared to neutral prompts (respectively 85% to 70% on average for Spanish and 75% to 50% for Bengali)
> Our interpretation is that this phenomenon is happening in English and French due to the distribution of formal/informal text in the pretraining data and the domains of the formal prompts. Indeed, we noticed that a lot of  neutral prompts result in formal text that follows unambiguously an encyclopedic style and topics, most likely due to pretraining on Wikipedia data. In comparison, the formal prompts used for French and English come from the XFORMAL dataset and are based on reformulated Yahoo answers. Even though formal, these prompts seem not to encode sufficient markers of formality to lead to formal outputs. We will discuss this in more detail in the paper. Thank you for your feedback on this.
>
> Answering your question about incohesiveness: as described in Appendix C, we force the model to generate at least 30 new subword tokens (excluding the prompt) to have a long enough generation sequence and be able to assess formality. The information provided l.446 is a mistake on our side, and we will fix it. The incohesive category only includes generations that are nonsense.
>
> Thank you for the typos and suggestions on formulation. We will also update the colors to be colorblind-safe.

---

### Official Review · Reviewer_emAM · 2023-08-04

**Soundness:** 3

**Excitement:**

3: Ambivalent: It has merits (e.g., it reports state-of-the-art results, the idea is nice), but there are key weaknesses (e.g., it describes incremental work), and it can significantly benefit from another round of revision. However, I won't object to accepting it if my co-reviewers champion it.

**Paper Topic And Main Contributions:**

The author's study the biases of generates text from two LLM's for formality across 5 languages. They discover some differences between the two models, as well as between languages which are interesting and worth further investigation. Furthermore, they release an annotated dataset of LLM generated output, which will be useful to the community at large towards studying formality in LLM output.

**Questions For The Authors:**

Questions are posed in point 2 of reasons to reject.

**Reasons To Accept:**

1. First formality study of LLM output in multilingual setting.
2. Interesting qualitative and quantitative analysis of LLM output in Section 5, especially in Table 4.

**Reasons To Reject:**

1. No discussion or analysis of inter-annotator agreement, which is understandable given that each output was only annotated by one native speaker. While it would be hard to do analysis over the entire corpus, it would be instructive to choose a subset of the corpus (10-50 outputs) and ask 2-3 native speakers to give formality ratings, followed by agreement statistics. Again, this might not change the results very much, but it would validate them, and individual annotator comments might be interesting in of themselves.

2. I'm not sure I agree with 5.2, where the authors suggest in the beginning that neutral prompts should lead to 50-50 distribution of formality. Shouldn't it lead to the distribution of formality in the pretraining corpus? No way to know for sure, but the sources for multilingual data might give an indication. Is it mostly online data (informal) or literature (probably formal)? In fact, I wonder if that is what Table 3 is showing. The authors note this in lines 521-533: perhaps a detailed analysis of the pretrained corpus and a comparison with the annotations would be interesting to add?

**Reproducibility:**

5: Could easily reproduce the results.

**Reviewer Confidence:**

4: Quite sure. I tried to check the important points carefully. It's unlikely, though conceivable, that I missed something that should affect my ratings.

---

> ### Author Rebuttal · Authors · 2023-08-28
>
> Thank you for your positive review and insightful comments.
>
> We agree with your feedback that collecting three ratings per sample on a subset of the corpus would ensure the quality of our annotation process. This project was done in a resource-constrained setting, and we could not onboard more annotators. Following your suggestion, we will collect two more ratings for at least 50 samples. We will include the inter-annotator agreement on this subset in our paper.
>
> Regarding your second point, our starting assumption (or a priori hypothesis) is that a language model used at scale should not be biased toward any level of formality. It is an assumption on the expected behavior of a “good” language model with regard to formality However, as we've noted in lines 521-533, we find that this hypothesis is disproved by our experiments. We explain the behavior by linking it to the pretraining data distribution. We will rephrase our assumption in line 475 to clarify the argumentation.

---

### Meta-Review · Area_Chair_DbpY · 2023-09-17

**Recommendation:** 3

**Metareview:**

This work is a novel contribution to studying the formality of LLMs' outputs in a multilingual setting. The authors provide an in-depth analysis of formality for five different languages and highlight the differences, which is important for characterizing and evaluating the formality of the model outputs. However, I agree with reviewer 13h5 that the definitions of informality are not clear and consistent across languages, which can potentially affect the evaluation of formality (and possibly the reproducibility of the work). This weakness is further emphasized by the fact that there is only one native speaker annotating the non-English outputs (as pointed out by reviewers 4rv1 and emAM).

Based on my personal reading of the paper, the reviewers' comments, and the post-rebuttal discussion, I believe that the setup is clear and well-designed to check the cohesion and formality variables. However, after AC-reviewers discussion, I think that the current version of the paper lacks some important in-depth analysis of the models' behaviors for more generalizable findings. For instance, how does the formality of text distributed in the pretraining corpora affect the formality-level bias and formality-level preservation? The paper briefly touches on this question (e.g., lines 521-533), but a careful empirical study is needed to determine whether the (aggregated) formality of text present in the pretraining corpora is a reasonably good predictor of the formality bias and preservation for any language. There must be (a few) causal factors that explain the contrasting behaviors across different model sizes and different languages. I don't think the current version of the paper can help answer "For a new language, what kind of outputs could we reasonably expect if we use a formal/neutral/informal prompt?" (in other words, generalizability of findings.)

---

### Decision · Program_Chairs · 2023-10-07

**Decision:**

Accept-Findings

**Comment:**

This work is a novel contribution to studying the formality of LLMs' outputs in a multilingual setting. The authors provide an in-depth analysis of formality for five different languages and highlight the differences, which is important for characterizing and evaluating the formality of the model outputs. However, I agree with reviewer 13h5 that the definitions of informality are not clear and consistent across languages, which can potentially affect the evaluation of formality (and possibly the reproducibility of the work). This weakness is further emphasized by the fact that there is only one native speaker annotating the non-English outputs (as pointed out by reviewers 4rv1 and emAM).

Based on my personal reading of the paper, the reviewers' comments, and the post-rebuttal discussion, I believe that the setup is clear and well-designed to check the cohesion and formality variables. However, after AC-reviewers discussion, I think that the current version of the paper lacks some important in-depth analysis of the models' behaviors for more generalizable findings. For instance, how does the formality of text distributed in the pretraining corpora affect the formality-level bias and formality-level preservation? The paper briefly touches on this question (e.g., lines 521-533), but a careful empirical study is needed to determine whether the (aggregated) formality of text present in the pretraining corpora is a reasonably good predictor of the formality bias and preservation for any language. There must be (a few) causal factors that explain the contrasting behaviors across different model sizes and different languages. I don't think the current version of the paper can help answer "For a new language, what kind of outputs could we reasonably expect if we use a formal/neutral/informal prompt?" (in other words, generalizability of findings.)